# Yield and Composition Variations of the Milk from Different Camel Breeds in Saudi Arabia

**Amr A. El-Hanafy** [1,2] , **Yasser M. Saad** [1,3] , **Saleh A. Alkarim** [1] , **Hussein A. Almehdar** [1] , **Fuad M. Alzahrani** [1] , **Mohammed A. Almatry** [1] , **Vladimir N. Uversky** [1,4,5,*] and **Elrashdy M. Redwan** [1,6,*]

1   Biological Sciences Department, Faculty of Sciences, King Abdulaziz University, P.O. Box 80203, Jeddah 21589, Saudi Arabia
2   Nucleic Acids Research Department, Genetic Engineering and Biotechnology Research Institute (GEBRI), City of Scientific Research and Technological Applications, New Borg El Arab, Alexandria 21934, Egypt
3   Genetics Laboratory, National Institute of Oceanography and Fisheries, Cairo 11516, Egypt
4   Institute for Biological Instrumentation of the Russian Academy of Sciences, Pushchino, Moscow Region 142290, Russia
5   Department of Molecular Medicine and USF Health Byrd Alzheimer's Research Institute, Morsani College of Medicine, University of South Florida, Tampa, FL 33612, USA
6   Therapeutic and Protective Proteins Laboratory, Protein Research Department, Genetic Engineering and Biotechnology Research Institute, City for Scientific Research and Technology Applications, New Borg El-Arab, Alexandria 21934, Egypt
*   Correspondence: vuversky@usf.edu (V.N.U.); lradwan@kau.edu.sa (E.M.R.)

**Abstract:** With the increasing interest in the identification of differences between camel breeds over the last decade, this study was conducted to estimate the variability of milk production and composition of four Saudi camel breeds during different seasons. Milk records were taken two days per week from females of Majahem, Safra, Wadha, and Hamra breeds distributed over Saudi Arabia. The milk yield during winter indicated that the weekly average of the Wadha breed was significantly lower (27.13 kg/week) than Majahem and Hamra breeds. The Safra breed had the lowest milk yield (30.7 kg/week) during summer. During winter, the Hamra breed had a lower content of all analyzed milk components except proteins and was characterized by a lower pH than the milk of the other breeds. However, the Hamra breed had significantly higher contents of milk fat and lactose than the other breeds during summer, with the corresponding values of 3.87 and 4.86%, respectively. Milk collected during winter from Majahem, Safra, and Wadha breeds was characterized by a significant increase in all milk components and milk pH. Finally, the isoelectric focusing analysis revealed noticeable variability of casein purified from camel milk within the different Saudi breeds, with the highest significant value of 2.29 g per 100 mL recorded for the Wadha breed.

**Keywords:** breed; camel; clan; milk composition; milk production; Saudi Arabia





## 1. Introduction

The camel is the principal domestic animal in Saudi Arabia, with its meat and milk representing a vital source of animal protein for nomads and city dwellers [1]. The total worldwide camel population is estimated to be 35 million head [2], with most of them being found in Africa. The greatest dairy camel population is present in Northeast African countries, such as Somalia, Ethiopia, and Sudan [3,4]. About 90% of the camels are one-humped (*Camelus dromedarius*), while 10% are two-humped (*Camelus bactrianus*) [5]. There is a continuous increase in the total number of camels used for milk production, mainly *Camelus dromedarius* [6], and this is accompanied by a noticeable increase in annual camel milk production [2]. In fact, in 2010 alone, about 5.25 million camels produced 2.12 million tons of milk [7]. Camels represent an important protein source for many humans in some parts of the world [8]. This is especially clear for the populations living in the arid parts of the world, such as nomadic communities, where other protein sources are scarce or almost

completely absent [9]. Furthermore, the need for various camel milk products has been on a constant rise over the last few years [2], and camel milk itself is gaining more attention as a healthy food [10]. The majority of the bioactive peptides found in dromedary camel milk came from caseins. Furthermore, glycosylation-dependent cell adhesion molecule 1 (GlyCAM-1) and peptidoglycan recognition protein 1 were discovered to be sources of ACE-inhibitory dipeptides VY and LY [11].

In the Arab world, the total dromedary population is estimated to be around 1.6 million head within the Arabian Peninsula, with 53% of these camels being found in Saudi Arabia [12,13]. Camels are highly valued in Saudi Arabia due to their adaptive characteristic, as they can survive in the hot and arid conditions predominant in Saudi Arabia. Camels are multipurpose animals, with females used mainly for milk production and males used primarily for transport or draft. In addition, both sexes produce meat as a tertiary product. A camel stores its energy reserves in the form of fatty depots in its body, especially in the hump and abdomen. For this reason, camels have the ability to survive for long periods without food and water [14].

Despite the major importance of camels as locally adapted livestock in Saudi Arabia, little information is available about the characteristic features of camel breeds. Most of the studies conducted rely upon morphological characterization and depend on coat color. There are several camel breeds identified in Saudi Arabia based on this parameter, e.g., Majahem, Wadih, Homor, Sofor, Shaele, Aouadi, Saheli, Awrk, Hadhana, Asail, Zargeh, and Shageh. However, the differences among breeds beyond this very general consideration are not clearly described. Ecological, morphological, and utilitarian criteria are generally mixed [15], and according to Al-Haknah, Saudi camel breeds are classified, taking into account their coat color, the purpose of use, and region of origin [16]. There is also some information about the difference in milk production among some Saudi camel breeds. In fact, several breeds of Saudi camels that are used for milk production, such as Majahem, Waddah, Sofor, Shaele, Zargeh, Komor, Shageh, and Awrk, are characterized by different milk yields, which are high for Sofor, medium to high for Komor and Shageh, and medium in Shaele, Zargeh, and Awrk clans. In addition, some breeds are specifically bred for meat production, such as Saheli and Hadhana. Other breeds are used in transport, work, and races, e.g., Aouadi and Hogen. A recent study examined the milk production of about 1400 female dromedaries with variable breeds (types), parities, sizes, and production potentials during the period from 2006 to 2009 in the United Arab Emirates [17]. The animals were raised under intensive conditions, and the authors reported an average daily milk production of about 6 kg (42 kg per week). They also reported an average length of lactation of about 586 days and stated that milk yield reached its peak during the 4th month after parturition. Several other studies were conducted to assess the milk yield and composition of camels in the Arab world and Saudi Arabia [17–19].

Similar to the milk of other species, camel milk contains two major classes of proteins: caseins and whey proteins. Caseins account for 80% ($w/w$) of the total protein content in camel milk [20]. The whey protein fraction contains numerous proteins, such as growth factors, immunoglobulins, $\alpha$-lactalbumin, lactoperoxidase, lysozyme, and lactoferrin [3]. Milk proteins serve very different biological functions. For example, lactoferrin is a multifunctional protein that plays a significant role in innate immunity and host defense against infection by microorganisms, alone or with other proteins such as lysozyme, lactoperoxidase, and immunoglobulins present in milk [21–24]. Lactoferrin is an iron-binding glycoprotein of the transferrin family that possesses antimicrobial activity. The content of lactoferrin in milk varies depending on the species [25]. Camel lactoferrin was shown to possess a superior antiviral activity as compared to human, bovine, sheep, and goat lactoferrins [26,27]. Camel immunoglobulins G (IgGs) have rather unexpected antiviral activities, which do not exist in other mammalian IgGs [28]. Lactoperoxidase is present in the milk of many species, where it oxidizes some organic and inorganic substrates with the catalytic production of hydrogen peroxide, thereby producing derivative compounds with antibacterial activity [29].

Due to its therapeutic and nutritional advantages, camel milk is now regarded as a symbol of health promotion. People who are allergic to cow's milk can use camel milk since it has several health-promoting qualities, including anti-adhesion and antibacterial characteristics. Furthermore, it was shown that camel milk did not induce allergies and also did not initiate diabetes mellitus, as reported for cow milk [30,31]. Furthermore, some data suggested that the daily consumption of camel milk might have a positive effect on diabetes patients [32–36]. Camel milk has small milk fat globules (MFGs), whose size ranges from 1.1 to 2.1 mm (note that MFGs found in buffalo, cow, and goat milk are noticeably larger, being 3.9–7.7 mm, 1.6–4.9 mm, and 1.1–3.9 mm, respectively). It is believed that the small size of camel MFGs contributes to the quicker digestion of camel milk [37]. Additionally, compared to cow milk, it contains less cholesterol and saturated fatty acids while also having higher levels of essential fatty acids, which results in an enhanced lipid profile and lower blood cholesterol levels [38]. Additionally, camel milk fat is high in phospholipids, particularly plasmalogens and sphingomyelin, which suggests that it may satisfy both adults' and children's daily nutritional needs [38]. It is also claimed that camel milk fat is a superior source of essential fatty acids (EFAs) [39], and in countries whose traditional diet is high in carbohydrates, consumption of camel milk might satisfy the daily nutritional needs of adults and infants in EFAs [38]. This is an important observation since in countries where the traditional diet is high in carbohydrate content, human milk can be characterized by low levels of EFAs, such as linoleic acid (LA), α-linolenic acid (ALA), and docosahexaenoic acid (DHA) [38,40,41]. It is recognized that compared to the milk of other mammalian species, camel milk serves as a good source of polyunsaturated FAs (PUFAs), such as ALA, eicosapentaenoic acid (EPA), and arachidonic acid (AA) [42–44]. This positions camel milk as a preferable source of fat for people with a high risk of lipid-related cardiovascular diseases [45,46] and type 2 diabetes [46]. It should be mentioned that although compared to cow milk, the average lipid content of mature camel milk is markedly lower, its cholesterol content is noticeably higher [47–50]. Paradoxically, several researchers have shown that the consumption of camel milk (in either fresh or fermented form) by rats can prevent the development of hypercholesterolemia [51–53].

Several genetic and nongenetic factors affect the chemical composition of camel milk. In this context, using a total of 1528 lactating dromedary camels, Nagy et al. [54] found that parity exerted a strong effect on all parameters of camel milk, where the primiparous dromedaries produced less milk with higher concentrations of components than did multiparous animals. Additionally, they reported that the stage of lactation and season strongly influenced milk yield and contents of all milk components. They also concluded that such seasonal variations were independent of nutrition and might reflect an endogenous circannual rhythm [54]. In another study, Nagy et al. [55] demonstrated variation in the major chemical composition of bulk dromedary camel milk using Fourier-transform mid-infrared (FT-MIR) spectroscopy. According to their findings, the yield of milk components and the chemical composition of bulk milk was strongly influenced by the month of the year, the study year, and the level of production [55]. However, the relative effect of season on composition was greater (proportion of variance app. 50%) than that of other factors of variation [55]. The importance of the laboratory analysis of raw milk before manufacturing has been emphasized [56]. As a lot may be learned about the quality of the products before their usage in subsequent manufacturing processes, raw milk and pasteurized milk are both examined in laboratories. Numerous factors, including animal husbandry, breeding procedures, the quality of feed, and the medical attention provided to the herd, affect the quality of milk, and these influences are amenable to evaluation by testing methods [56]. The importance of such analysis was raised primarily because of the presence of specific allergens in milk, including casein and lactose [56].

In this work, we conduct a systematic analysis of seasonal variability in the yield and composition of camel milk collected in northern Jeddah, Riyadh, and Alwagh governorates of Saudi Arabia from Majahem, Safra, Wadha, and Hamra breeds.

## 2. Materials and Methods

### 2.1. Animals and Locations

This study was carried out from June 2015 to December 2017 and used 60 female camels of four breeds/clans, which were enrolled in this study over four seasons/one year. The camel herd used in this study included four ecotypes, namely Majahem, Safra, Wadha, and Hamra. Their appearance is shown in Figure S1.

Selected camels were in the second to third lactation season, and samples were collected starting from 60 days post-partum. The camels were kept indoors and outdoors, and their diet over the year was mainly composed of alfalfa, *Ammophila arenaria*, *Hordeum vulgare*, and *Sorghum bicolor*, and barley, wheat bran was used as feed supplements according to nutritional requirements, especially in winter.

The animals were located in different private farms in northern Jeddah (16 female, four from each ecotype), Riyadh (28 female, 6 from each ecotype), and Alwagh (16 female, four from each ecotype) governorates, Saudi Arabia (Figure 1), covering a large part of Saudi Arabia (area ~500.000 Km, Figure 1).

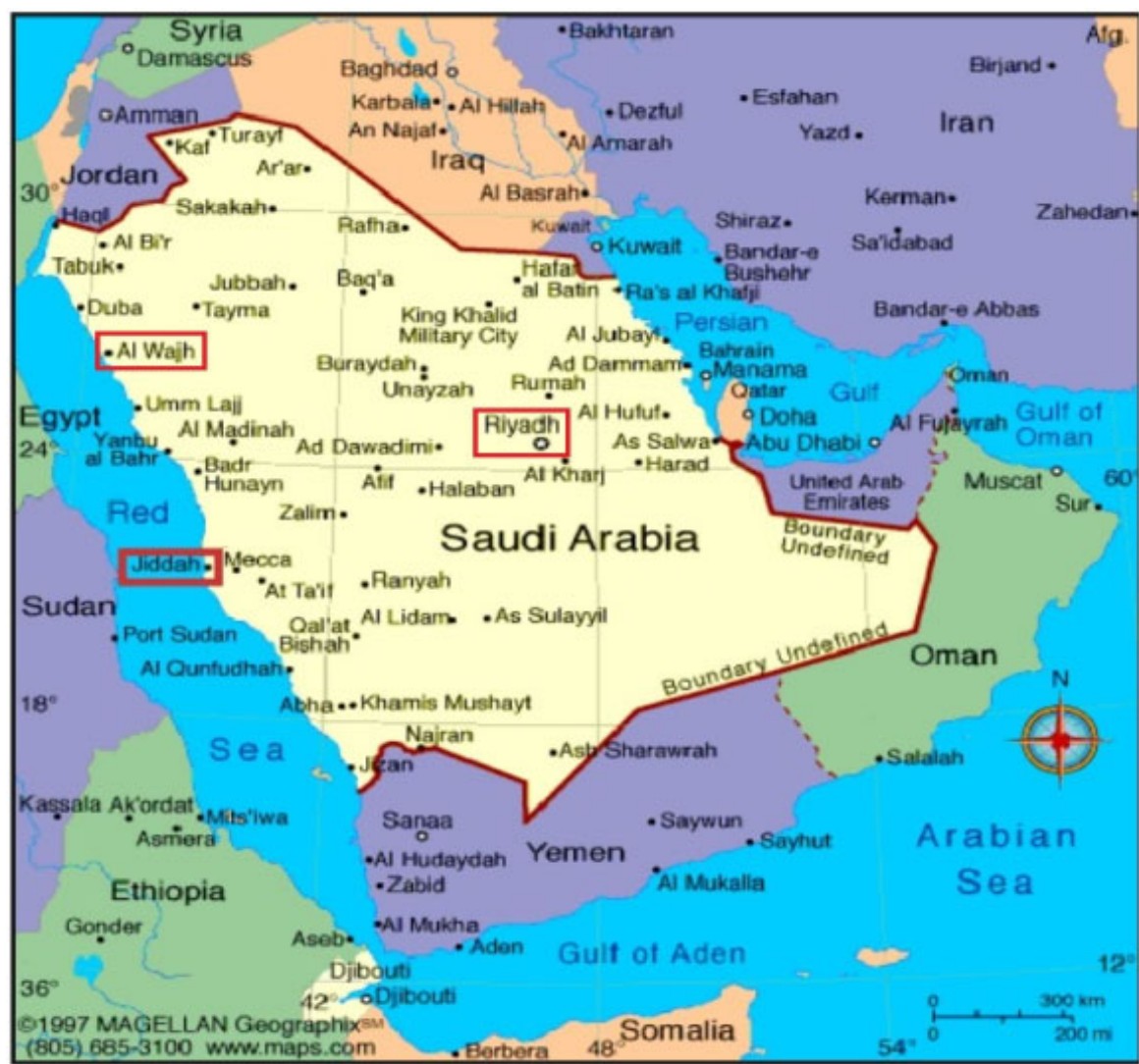

**Figure 1.** Map of Saudi Arabia, with red boxes pointing to the locations where camel milk samples were collected from private farms.

The annual average temperature was 28 °C, ranging from 15–40 °C, according to the general authority of meteorology and environmental protection (https://www.pme.gov.

sa/en/pages/default.aspx, accessed on 10 March 2018). The following parameters were recorded for each female camel used: parity (primiparous or multiparous), gestation length, and age. The related information on the camels was provided by educated farm directors, who were either veterinary doctors or specialists with bachelor's degrees. There was no difference in the feeding regime or management between farms.

## 2.2. Milk Yield

Milk samples were collected on private farms in the early morning (5:00–6:00 a.m.) into clean plastic bottles (100–300 mL) in four locations spread over the northern Jeddah, Riyadh, and Alwagh governorates of Saudi Arabia, often before the manual milking process. After good mixing, milk samples were coded, and a 0.1% sodium azide was added, then immediately kept frozen till transferred to our lab. The milk samples were collected from female camels of different breeds and gestation. Milk records were taken two days in each week on the same day to get a precise 7-day interval, and the average weekly yields were calculated. Average weekly yields from different camel breeds were statistically analyzed.

## 2.3. Milk Composition

All milk samples collected were analyzed, and the arithmetic means were calculated. The means were used as a single value for the statistical analysis of each animal over four ecotypes and seasons. Fifteen animals (four replicates per animal) were used per clan to assess milk parameters. Milk contents of fat, protein, lactose, solids-not-fat (SNF), salts, and pH were analyzed at room temperature using automatic Lactoscan (MCC WS®, Nova Zagora, Bulgaria, https://www.lactoscan.com/, accessed on 10 March 2018). Analysis was conducted according to the manufacturer manual after the Lactoscan was calibrated for the camel milk with the help of the company programmer. Data for each parameter of milk composition for different breeds, both within each season and over the entire period of observation, were statistically analyzed. As per the Lactoscan manufacturer manual, the content of minerals is defined as Salts = SNF × MSCC, where SNF is solids-not-fat in percentage, and MSCC is a milk-specific constant coefficient, which is equal to 0.083% or 0.075% for cow and sheep milk, respectively).

## 2.4. Purification and Electrophoresis of Casein from Different Camel Breeds

Pooling milk samples from camels within each breed/each season, then 100 mL from the pooled milk were used to prepare cream and casein. Camel milk samples from four breeds were skimmed according to the procedure described in [57]. The produced cream layers were washed several times with cold phosphate buffer saline (PBS) and then weighed out. The pH of skimmed milk was lowered to pH 4.6; then, the samples were centrifuged to precipitate casein. These casein-containing samples were washed with cold Tris-HCl buffer (pH 4.6) several times. After the last washing and centrifugation, the purified caseins of different camel breeds were analyzed by SDS-PAGE and SDS-Urea-PAGE [58] and then lyophilized. Casein fractionation of four camel clans based on the isoelectric focusing was conducted according to the standard protocols described earlier [59,60].

## 2.5. Statistical Analysis

The averages of the weekly milk yield and composition from different camel breeds were statistically analyzed using ANOVA single factor, and the group differences were determined with the Fisher Least Significant Difference (LSD) Method, with probability at $p < 0.05$ considered significant.

## 3. Results and Discussion

### 3.1. Analysis of the Seasonal Variation in Milk Yield

Camel milk has developed a high reputation as a healthy nutrient, with most of its therapeutic value ascribed to its biological properties, so exploring camel milk production and composition should be considered a priority. Table 1 represents the overall means of



the weekly milk yields of various camel clans during different seasons. The average weekly milk yield of the Safra camel breed (30.05 kg/week) was significantly ($p < 0.05$) lower than the yields of the other three breeds, which ranged from 39.68 kg/week for Hamra to 42.42 kg/week for Majahem, while there were no significant differences between these three breeds.

**Table 1.** Average milk yield (kg/week ± S.E.) of Saudi camel breeds during various seasons.

| Breeds | Milk Yield (kg/Week) | | | |
|---|---|---|---|---|
| | Spring | Summer | Autumn | Winter |
| Majahem | 45.0 ± 4.1 [a] | 41.9 ± 2.4 [a] | 31.33 ± 0.77 [a] | 34.0 ± 3.1 [a] |
| Safra | 39.3 ± 5.1 [b] | 30.7 ± 3.4 [b] | 30.07 ± 0.85 [a] | 31.1 ± 6.2 [a,b] |
| Wadha | 41.9 ± 5.6 [a,b] | 39.1 ± 2.0 [a] | 25.3 ± 1.2 [b] | 27.1 ± 4.6 [b] |
| Hamra | 39.9 ± 4.7 [b] | 40.1 ± 1.3 [a] | 30.5 ± 0.52 [a] | 34.7 ± 6.6 [a] |

The means within each column with different superscripts ([a,b]) are statistically different ($p < 0.05$).

It is also clear from Table 1 that the Wadha breed had the lowest milk yield ($p < 0.05$) during the summer season (25.25 kg/week). However, during summer, there were no significant differences between the other three breeds, which ranged from 31.33 kg/week for the Majahem breed to 30.07 kg/week for the Safra breed. Additionally, there was an increase in milk yield during the autumn season for all breeds studied, except for Safra, when compared to the milk yield during the summer season. Milk yield during autumn obtained in the current study is in accordance with what was reported for Maghrebi female camels raised in the Marsa Matrouh Governorate of Northwest Egypt [18]. In fact, these authors reported milk yields ranging from 28 to 39.9 kg/week, depending upon parity, where the highest milk yield was recorded between the fifth and eighth parities, while the lowest value was recorded at the first and second parities [18].

Nagy et al. analyzed data from about 1400 female dromedaries with variable breeds (types), parities, sizes, and production potentials during the period from 2006 to 2009 in the United Arab Emirates [17]. These animals were raised under intensive conditions, and the data on this farm were collected during the first three years of operation. These authors reported an average daily milk production of about 6 kg (42 kg/week), which is almost identical to what was obtained in the current study for the Majahem breed and close to what was seen for the Wadha and Hamra breeds. The authors reported an average length of lactation of about 586 days and also stated that milk yield reached its peak during the 4th month after parturition [17]. The increase in milk yield obtained in this study during the autumn season, when compared to the summer season, follows the results reported in [61]. Those authors analyzed data from 47 female camels belonging to different local breeds (Majaheem, Waddah, and Homor). The herd was under intensive feeding management, and the animals were housed in open-air shade pens. It was found that camels that calved during the cold months (November to February) were the most productive, with the highest persistency, peak yield, and longest lactation length [61]. The reduction in the milk yield observed in the current study may be attributed to the increase in sweating and heat stress during the hot weather of the summer season.

Table 1 shows that the average weekly milk yield of the Wadha camel breed during the winter season (27.13 kg/week) was significantly ($p < 0.05$) lower than the milk yields of the Majahem and Hamra breeds.

Although the Safra breed had higher milk production, the differences in the yield levels were not significant when compared to the Wadha breed. Additionally, there were no significant differences between Majahem, Safra, and Hamra breeds during the winter season. During summer, the milk yield values of the four breeds were close to each other, with the Safra breed showing the lowest yield. During the spring season, values obtained were close to those of the autumn season, where the Majahem breed showed the highest significant levels ($p < 0.05$) (45 kg/week) when compared to the milk yields of the

Safra and Hamra breeds, though the difference between Majahem and Wadha breeds was less pronounced.

### 3.2. Seasonable Variability in the Milk Composition of Saudi Camel Breeds

Table 2 represents the overall means of different milk components and milk pH for different Saudi camel breeds over four seasons. It can be concluded from the presented data that, except for milk fat and pH, there were no significant differences among the four breeds studied.

On the other hand, milk fat and pH were significantly higher ($p < 0.05$) in the Majahem and Safra breeds than in the Wadha and Hamra breeds. Table 2 also shows that during winter, the Hamra breed had significantly lower ($p < 0.05$) percentages of all milk components (except the protein level) and lower pH than the other breeds. When examining the milk composition during the summer season, it is clear that the Hamra breed had significantly higher ($p < 0.05$) milk fat and lactose contents than the other three breeds, which, however, did not show significant differences among each other. Additionally, the Hamra and Safra breeds had significantly higher ($p < 0.05$) milk protein content, salt levels, and pH values than the Majahem and Wadha breeds. No significant differences between the milk solids-non-fat (SNF) content were noted among the four studied breeds. It is also worth noting that there was an increase in all milk components during the winter season for all breeds except Hamra when compared to those evaluated during the summer season.

The average content of milk fat obtained in the current study for the four breeds, which ranged from the lowest value of 2.14% for the Hamra breed to the highest value of 3.07% observed for the Majahem breed, agrees with the results obtained by Mehaia et al., who reported that milk fat content in the Majahem breed was about 3.22% [62]. As for the Wadha breed, these authors reported values (2.46%) almost identical to what was obtained in the current study [62]. However, they reported a higher value for Hamra (2.85%) when compared to our value of 2.14%.

Additionally, they reported lactose contents for these three breeds that were comparable to those found in the current study. These authors also reported lower milk protein contents for Wadha and Hamra (2.36 and 2.52%, respectively) than observed in the current study, while the milk of the Majahem breed contained 2.91% proteins, which is very close to our value (2.89%) [62].

Abdalla et al. mentioned that the milk fat content in the Maghrebi female camels raised in the Marsa Matrouh Governorate of Northwest Egypt ranged from 2.5% to 3.7%, depending on the time of the lactation season, with the highest value obtained during the peak of milk production [18]. These authors reported that the calculated means of milk components were obtained based on the analysis of data from 748 records of 43 Maghrebi camels during 73 lactations. The lower overall milk fat content of Wadha and Hamra breeds could also be explained based on the higher milk yield of these two breeds when compared to the yields of the other two breeds analyzed in the current study. Additionally, these authors reported values of protein, lactose, and ash contents (3.01%, 4.33%, and 0.69%, respectively) that were very similar to the corresponding values found in our study [18]. The notable increase in milk fat, protein, lactose, and salt content during winter compared to summer for Saudi camel breeds studied here agrees with the results obtained by Haddaddin et al. [63]. Their study examined camels from an area called Dair Al Quin (located in the northeast of Jordan), about 170 km from the capital, Amman. They reported that the large seasonal variation was in total solids and fat contents, where the maximal values were observed in mid-winter (13.9% and 3.9%), and the minimum values were obtained in August (10.2% and 2.5% for total solids and fat contents, respectively).

**Table 2.** Variability of milk composition (mean ± SE) of different Saudi camel breeds in summer, winter, spring, and autumn seasons over 12 months.

| | Saudi Camel Clan Milk Composition | | | | | | | | | | | |
|---|---|---|---|---|---|---|---|---|---|---|---|---|
| **Breed** | **Fat%** | | | | **Protein%** | | | | **Lactose%** | | | |
| | **Spring** | **Summer** | **Autumn** | **Winter** | **Spring** | **Summer** | **Autumn** | **Winter** | **Spring** | **Summer** | **Autumn** | **Winter** |
| Majahem | 3.65 ± 0.34 | 1.91 ± 0.06 [b] | 2.81 ± 0.52 | 3.47 ± 0.18 [a] | 3.15 ± 0.04 [a] | 2.61 ± 0.05 [b] | 2.72 ± 0.08 | 3.10 ± 0.03 | 4.73 ± 0.06 [a,d] | 3.92 ± 0.07 [b] | 4.08 ± 0.12 | 4.65±0.04 [a] |
| Safra | 4.00 ± 0.12 | 1.77 ± 0.20 [b] | 2.68 ± 0.62 | 3.91 ± 0.41 [a] | 2.93 ± 0.05 [b] | 2.90 ± 0.09 [a] | 2.76 ± 0.25 | 3.05 ± 0.06 | 4.40 ± 0.07 [b,c] | 4.23 ± 0.04 [b] | 4.14 ± 0.37 | 4.63 ± 0.06 [a] |
| Wadha | 3.78 ± 0.27 | 2.26 ± 0.11 [b] | 1.53 ± 0.22 | 3.43 ± 0.18 [a] | 3.18 ± 0.05 [a] | 2.67 ± 0.10 [b] | 2.78 ± 0.10 | 3.12 ± 0.09 | 4.77 ± 0.07 [d] | 3.89 ± 0.11 [b] | 4.18 ± 0.15 | 4.67 ± 0.07 [a] |
| Hamra | 4.04 ± 0.14 | 3.87 ± 0.26 [a] | 2.47 ± 0.45 | 2.25 ± 0.23 [b] | 3.05 ± 0.05 [a,b] | 2.95 ± 0.02 [a] | 2.89 ± 0.09 | 2.93 ± 0.04 | 4.56 ± 0.04 [a,c] | 4.64 ± 0.20 [a] | 4.35 ± 0.14 | 4.40 ± 0.06 [b] |
| | Saudi Camel Clans' Milk Composition | | | | | | | | | | | |
| **Breed** | **Salt%** | | | | **SNF%** | | | | **pH** | | | |
| | **Spring** | **Summer** | **Autumn** | **Winter** | **Spring** | **Summer** | **Autumn** | **Winter** | **Spring** | **Summer** | **Autumn** | **Winter** |
| Majahem | 0.71 ± 0.01 [a,d] | 0.58 ± 0.01 [b] | 0.61 ± 0.01 | 0.70 ± 0.01 [a] | 8.60 ± 0.11 [a] | 7.14 ± 0.13 | 7.43 ± 0.23 | 8.74 ± 0.08 [a] | 6.71 ± 0.015 [a] | 6.37 ± 0.02 [b] | 6.71 ± 0.07 | 6.83 ± 0.01 [a] |
| Safra | 0.66 ± 0.11 [b,c] | 0.63 ± 0.01 [a] | 0.63 ± 0.05 | 0.69 ± 0.01 [a] | 8.00 ± 0.14 [b] | 7.70 ± 0.08 | 7.53 ± 0.68 | 8.42 ± 0.12 [a] | 6.61 ± 0.023 [b] | 6.50 ± 0.01 [a] | 7.01 ± 0.10 | 6.82 ± 0.02 [a] |
| Wadha | 0.71 ± 0.01 [d] | 0.58 ± 0.01 [b] | 0.62 ± 0.02 | 0.70 ± 0.011 [a] | 8.55 ± 0.19 [a] | 7.08 ± 0.20 | 7.60 ± 0.28 | 8.52 ± 0.14 [a] | 6.79 ± 0.021 [c] | 6.40 ± 0.02 [b] | 6.49 ± 0.09 | 6.87 ± 0.03 [a] |
| Hamra | 0.68 ± 0.01 [a,c] | 0.66 ± 0.01 [a] | 0.65 ± 0.02 | 0.66 ± 0.01 [b] | 8.33 ± 0.08 [a,b] | 7.70 ± 0.39 | 7.88 ± 0.24 | 7.97 ± 0.11 [b] | 6.77 ± 0.02 [c,d] | 6.48 ± 0.01 [a] | 6.50 ± 0.15 | 6.72 ± 0.28 [b] |

The means within each column with different superscripts ([a,b,c,d]) are statistically different ($p < 0.05$).

Haddaddin et al. also reported that the lactating camels were fed on natural desert plants throughout the year except for winter, when each evening, animals were supplemented with dried barley [63]. Therefore, the enhancement of the fat and total solid contents in their study could be attributed to the additional feed allowances provided during the winter season. Additionally, the enhancement in milk quality observed in the current study could be explained by the improvement in pasture conditions during the winter season in Saudi Arabia and additional feed supplements, which could be provided during this season. In another study, Musaad et al. reported seasonal variation in milk fat, protein, and lactose contents in four Saudi ecotype breeds, Malhah, Wadha, Hamra, and Safra [19]. In this study, it was mentioned that the maximal fat content was observed in January (3.46%), and the minimal fat levels were recorded during July (2.29%). The same trend was noted for protein and lactose contents, while rather small seasonal changes were observed in milk ash content [19].

Table 2 also represents data for the milk compositions of different Saudi camel breeds during the four seasons. It is obvious from these data that there were no significant differences between the four breeds during these seasons with respect to the milk fat content, except for Wadha, which had the lowest value. In addition, milk protein and SNF contents followed the same pattern, where there were significant differences between levels of both components in the Majahem, Wadha, and Safra breeds in favor of the Majahem and Wadha breeds, while there were no significant differences between the values obtained for the Majahem and Wadha breeds or for the Majahem and Wadha breeds when compared to the Hamra breed. Additionally, there was no significant difference between Safra and Hamra breeds. On the other hand, milk lactose and salt levels followed the previously described pattern, except that there was a significant difference between values measured for the Wadha and Hamra breeds in favor of the Wadha breed. There were significant differences in milk pH between the four Saudi camel breeds except for Wadha and Hamra breeds. The highest pH value was recorded for the Safra breed, whereas the lowest value was that of the Wadha breed.

A systematic review and meta-analysis that included a total of 7298 camel milk samples from 23 countries were conducted by Alhaj et al. in 2022 [64]. They examined 79 reports written in English and published in 1980 or later, as well as 117 assessments of seasons, sub-breeds, and nations [64]. Based on this analysis, it was concluded that differences in camel milk profiles depended on multiple variables, including the number of studies and samples included, the analytical techniques used, the feeding habits, the breeds of camels, the geographic location, and the seasons [64]. Hanganu et al. pointed out that butyric acid content may be used as a marker to distinguish among fats of dairy or non-dairy origin, as well as to distinguish milk fats from different species [65]. Based on these and related observations, the analysis of camel milk short-chain fatty acids was recommended for future studies to detect adulterated products as well as to distinguish among milk fats from different species [66].

### 3.3. Fractionation of Camel Casein and Cream

Table 3 demonstrates the camel milk cream content and casein concentration for each clan. Safra milk was characterized by the highest cream value, whereas the lowest value was found in Wadha milk over four seasons. However, the highest casein concentration was in Wadha milk, and the lowest concentration was seen in the Safra breed over four seasons. These concentrations are similar to those previously reported [67], though they are slightly higher than those found for Majahem, Hamra, and Wadha breeds [62].

Figure 2 shows that on a normal, reducing SDS gel (see Figure 2A), milk proteins migrated as expected, and the positions of different protein bands mostly coincided with previously published data [5,8,68–70]. On the other hand, Figure 2B illustrates that, in addition to the increase in the resolution of protein bands, the urea-SDS gel affected protein migration, and proteins were positioned differently in comparison with normal SDS-PAGE.

**Table 3.** Total milk cream and casein (gm) contents (Mean ± SE) in Saudi camel breeds (Safra, Wadha, Hamra, Majahem).

| | From 100 mL of Camel Milk | | | |
| --- | --- | --- | --- | --- |
| | **Safra** | **Wadha** | **Hamra** | **Majahem** |
| **Cream** | 6.76 ± 1.09 [a] | 2.69 ± 0.76 [b] | 3.33 ± 0.87 [b] | 4.35 ± 0.73 [b] |
| **Casein** | 1.73 ± 0.45 [b] | 2.29 ± 0.56 [a] | 2.07 ± 0.42 | 1.86 ± 0.39 [b] |

The means within each row with different superscripts ([a,b]) differ ($p < 0.001$) for cream and ($p < 0.05$) casein. The average represents four breeds over four seasons.

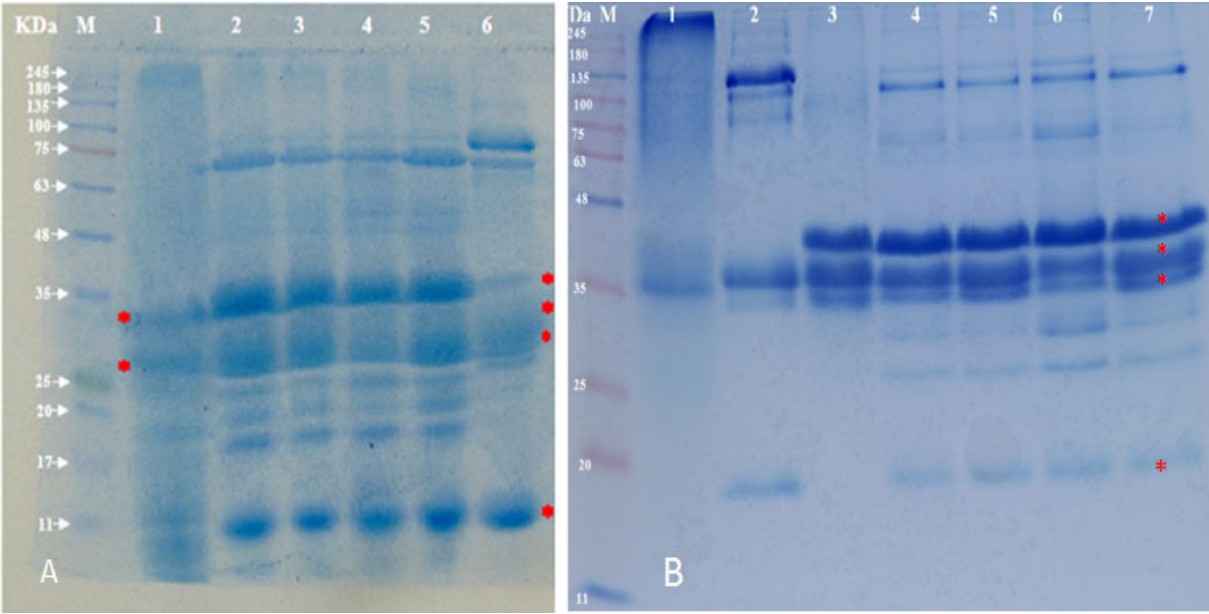

**Figure 2.** Analysis of milk proteins using 12.5% Sodium dodecyl sulfate-polyacrylamide gel electrophoresis (**A**) and Sodium dodecyl sulphate-Urea polyacrylamide gel electrophoresis (**B**). The panel A profile includes purified bovine casein (Sigma-Aldrich C-4765; Merck KGaA, Darmstadt, Germany) in lane 1, camel milk proteins from Wadha, Hamra, Safra, and Majahem clans (lanes 2, 3, 4, and 5, respectively), and human milk proteins in lane 6. Panel B includes purified bovine casein (Sigma-Aldrich C-4765) in lane 1 and human milk proteins in lane 2, and camel milk proteins from Wadha, Hamra, Safra, and Majahem clans (lanes 3, 4, 5, and 6, respectively), and lane 7 is the repeated analysis of Majahem milk. The leftmost lane in both panels shows the pre-stained molecular mass standard proteins. The red asterisks from up to down are pointed at the casein bands, then the camel and human α-lactalbumin bands.

In fact, although the molecular weights of camel casein fraction (α-, ß- and k-casein, respectively) ranged from 35 to 23 kDa in SDS-PAGE (Figure 2A), they migrated slower in urea-SDS-PAGE (see Figure 2B), and their apparent molecular masses ranged from 40 to 35 kDa. The band corresponding to k-casein in these gels was the smallest in terms of both molecular mass and concentration [71]. In our study, the content of casein fraction in the Safra breed was the lowest, which was confirmed by data shown in Table 3 and Figure 2. Additionally, α-lactalbumin migrated similarly to casein in both gels. The small differences in the migration of human and camel α-lactalbumin, seen in both Figure 2A,B gels, may be due to the differences in their amino acid sequences [72].

Results of the isoelectric focusing (IEF) analysis of the purified caseins from different Saudi camel breeds are shown in Figure 3. The casein protein profiles in these gels were noticeably different from those shown in Figure 2, with five discrete bands of high intensity being clearly seen in Figure 3 and with two additional fine bands also being present. There were differences in the IEF profiles of different camel clans. The red asterisk points to

the protein band found in the Wadha, Safra, and Hamra clans but not expressed in the Majahem breed, whereas the yellow asterisk points to the protein band missing in the Wadha clan but expressed in all other clans (Figure 3, lane 2).

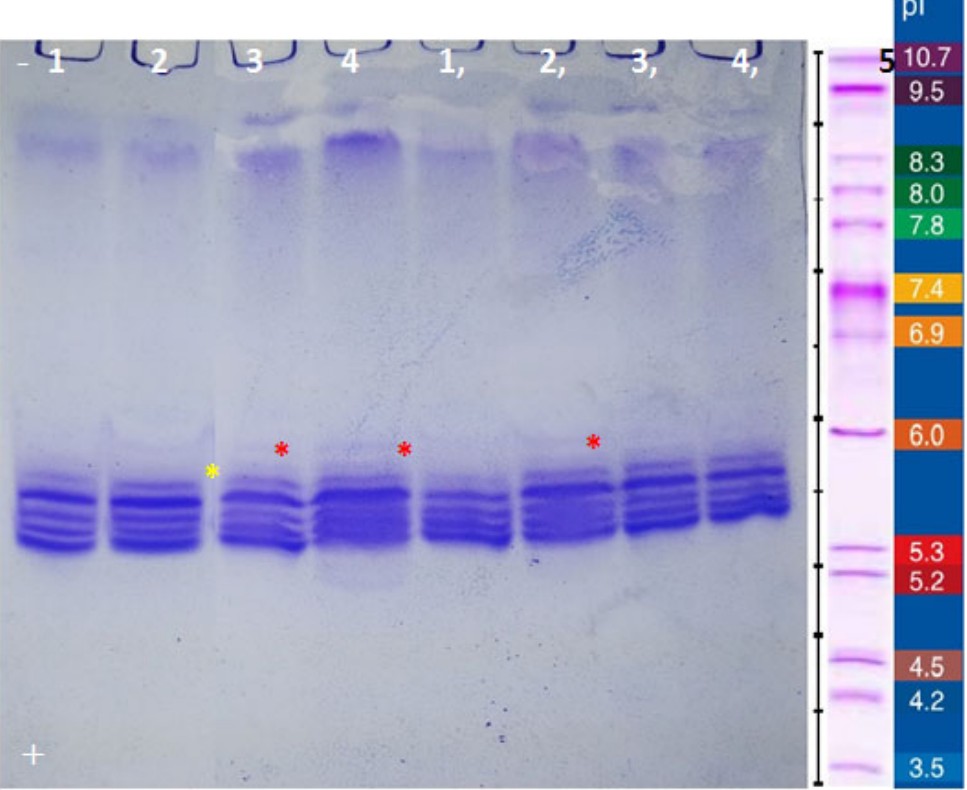

**Figure 3.** Profiling of casein purified from camel milk according to its isoelectric focusing (IEF) in slab gel of pH range 3–10. Lane 5 and the right bar represent Serva IEF marker, lanes 1–4 show casein from the milk of Saudi camel clans Majahem, Wadha, Hamra, and Safra, lanes 1′–4′, same camel clans sequences but with new individual milk, respectively. Plus and minus signs indicate the anode and cathode electrodes, while red and yellow asterisks indicate the existence (a faint band) and mission IEF band, respectively.

Although these observations may indicate the presence of some inter- and intra-clan genetic variability in Saudi camel clans, these data are too preliminary and require additional forthcoming analysis. Our data seem to agree with the results of Erhardt et al. [73] and Redwan et al. [74] regarding the analysis of camel milk collected from various Sudan and Saudi Arabia breeds, which revealed the existence of several protein variants in camel milk collected from different locations over Sudan and Saudi Arabia [73].

It is known that there are four main types of milk caseins, $\alpha_{s1}$-casein, $\alpha_{s2}$-casein, $\beta$-casein, and $\kappa$-casein, encoded by *CSN1S1*, *CSN1S2*, *CSN2*, and *CSN3* genes, respectively. On average, in camel milk, $\alpha$-casein (22%) is the second largest percentage after $\beta$-casein (65%) [3,8]. Curiously, our analysis showed that not only were global casein contents different in milk samples of different breeds of Saudi camels, but the relative levels of different caseins varied among the breeds as well. Milk protein composition traits are associated with protein genetic variants. For example, camel $\alpha$-casein may exist as several variants. This follows from the elegant work of Erhardt et al., who analyzed casein gene polymorphism in milk samples of camels from different regions of Sudan (Africa) using isoelectric focusing [73]. This analysis revealed that at least three different variants, A, C, and D, could be identified for $\alpha$-casein. In the different ecotypes, the major allele A was present with frequencies of 0.79 (Lahaoi), 0.75 (Shanbali), 0.90 (Arabi Khali), and 0.88 (Arabi Gharbawi) [73]. Similarly, our previous analysis of milk proteome and DNA sequences

of both α-lactalbumin and casein revealed low levels of genetic differences among and within the Saudi camel clans [74]. Furthermore, in comparison to the α-caseins A and D, the C variant of camel α-casein showed a single G > T nucleotide substitution in the exon 5, leading to a non-synonymous amino acid substitution (p.Glu30 > Asp30, GenBank ID: JF429138) [73].

Camel milk has been recently shown by Mudgil et al. to serve as an excellent source to obtain hypoallergenic infant milk formulae, and consequently, it should be considered a good alternative to cow-milk-based products [75]. The formulation, characteristics, and in vitro digestion of the novel camel milk-based infant formula were compared to those of the commercial and bovine infant formulas. Based on these analyses, the authors concluded that camel milk could be a viable alternative milk source for newborn formula manufacturing [75]. The protein digestibility pattern of camel milk-based infant formula was comparable to that of commercial and bovine milk-based infant formulas. Additionally, camel whey or casein fortifications considerably increased the high radical scavenging activity of the camel milk-based infant formula as compared to those of the commercial infant formula [75]. Infant formula made from digested camel milk had anti-inflammatory properties that were superior to those of the commercial infant formula and bovine milk, indicating that it would be a feasible option for the creation of a hypoallergenic formula for infants who are allergic to bovine milk [75,76]. Additionally, peptic camel whey hydrolysates were examined as potential sources of peptides with antihypertensive properties capable of inhibiting the angiotensin-converting enzyme and renin [77]. The authors concluded that this antihypertensive potential was significantly influenced by different hydrolysis conditions used to produce camel whey hydrolysates. The significance of the hydrophobic amino acids, especially proline, in antihypertensive peptides is further supported by this study [77].

Obviously, studies similar to the one reported here are needed for different camel breeds found in Saudi Arabia. However, we hope that our results can be considered a starting point for future characterization of the genetic diversity of camel milk proteins. Related information can also be used for establishing the association between milk protein variability and milk performance traits in camel clans distributed over Saudi Arabia.

## 4. Conclusions

Results of our analyses of seasonal variation in milk yield and composition clearly showed that the Safra and Wadha breeds had significantly lower milk yield during winter and summer. There also were significant differences in milk components among the four Saudi camel breeds during winter, summer, and spring seasons, with the tendency to increase the levels of milk components during the winter season. Furthermore, isoelectric focusing revealed the presence of noticeable variability in the contents of different camel milk casein types between different Saudi breeds. Although presented data indicate the presence of noticeable variability in milk yield and composition among the Saudi camel clans, further in-depth analysis is needed to better understand seasonal and between-clan variability of camel milk production and composition, as well as the presence of changes in the milk proteome.

**Supplementary Materials:** The following supporting information can be downloaded at: https://www.mdpi.com/article/10.3390/sci5010002/s1, Figure S1. Ecotype of Saudi camel clans (Hamra, Safra, Majahem, and Wadha), the milk of which was analyzed in this study.

**Author Contributions:** Conceptualization, A.A.E.-H., Y.M.S., S.A.A., H.A.A., V.N.U. and E.M.R.; methodology, M.A.A., Y.M.S., S.A.A., H.A.A., V.N.U. and E.M.R.; formal analysis, A.A.E.-H., M.A.A., Y.M.S. and S.A.A.; investigation, A.A.E.-H., Y.M.S., S.A.A., H.A.A., F.M.A. and M.A.A.; validation, A.A.E.-H., Y.M.S., S.A.A., H.A.A., F.M.A., M.A.A., V.N.U. and E.M.R.; writing—original draft preparation, A.A.E.-H., Y.M.S., S.A.A., H.A.A., F.M.A., M.A.A., V.N.U. and E.M.R.; writing—review and editing, V.N.U. and E.M.R.; supervision, E.M.R.; project administration, E.M.R. and H.A.A. All authors have read and agreed to the published version of the manuscript.

**Funding:** This work was supported by a grant from the King Abdulaziz City for Science and Technology (A.B-35-195).

**Institutional Review Board Statement:** Not applicable.

**Informed Consent Statement:** Not applicable.

**Data Availability Statement:** Not applicable.

**Conflicts of Interest:** The authors declare no conflict of interest. The funders had no role in the design of the study; in the collection, analyses, or interpretation of data; in the writing of the manuscript; or in the decision to publish the results.

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
