# Peer review of "Yield and Composition Variations of the Milk from Different Camel Breeds in Saudi Arabia"

_sci, doi:10.3390/sci5010002_

Round 1

Reviewer 1 Report

This manuscript estimates camel milk production and composition variability from different breeds during different seasons. This manuscript could be reconsidered for publication after conducting a major revision in the introduction and discussion parts. I am hereby returning the paper to you, pointing out comments to the authors:

Abstract:

L21, remove with

Introduction

L37, remove the introduction word due to duplication.

The authors did not report all factors that affect camel milk composition; please refer to the below articles to support the introduction.

·         Effect of genetic and nongenetic factors on chemical composition of individual milk samples from dromedary camels (Camelus dromedarius) under intensive management. Journal of Dairy Science, 100(11), 8680–8693. https://doi.org/10.3168/JDS.2017-12814

·         Circannual changes in major chemical composition of bulk dromedary camel milk as determined by FT-MIR spectroscopy, and factors of variation. Food Chemistry, 278, 248–253. https://doi.org/10.1016/J.FOODCHEM.2018.11.059

·         Physico-chemical quality of camel milk. Journal of Agriculture and Social Sciences, 2, 164–166.

·         Laboratory analysis of milk and dairy products. In R. C. Chandan, A. Kilara, & N. P. Shah (Eds.), Dairy processing and quality assurance (pp. 600–646). John Wiley & Sons. https://doi.org/10.1002/9781118810279.CH24

Materials and Methods

L 110-111, Add stage of lactation  

Results and Discussion

In general, results are only presented in the main text without proper discussion; you can add the below article and factors to support the discussion.

L176, add discussion

Section 3.2, support this section with the most recent meta-analysis study related to the effect of season, breed and country on camel milk composition

Camel milk composition by breed, season, publication year, and country: A global systematic review, meta-analysis, and meta-regression. Compr Rev Food Sci Food Saf. 2022 May;21(3):2520-2559.

Add to the discussion part, variations in camel milk composition are also due to several factors, including the number of samples, different analytical techniques, and livestock management.

 Throughout the manuscript, the use of the English language needs to be improved.

Author Response

Reviewer #1

This manuscript estimates camel milk production and composition variability from different breeds during different seasons. This manuscript could be reconsidered for publication after conducting a major revision in the introduction and discussion parts. I am hereby returning the paper to you, pointing out comments to the authors:

Abstract:

L21, remove with

Reply: removed

Introduction

L37, remove the introduction word due to duplication.

Reply: removed

The authors did not report all factors that affect camel milk composition; please refer to the below articles to support the introduction.

Reply: This information and corresponding references were added to the revised manuscript (lines 135-155)

  • Effect of genetic and nongenetic factors on chemical composition of individual milk samples from dromedary camels (Camelus dromedarius) under intensive management. Journal of Dairy Science, 100(11), 8680–8693. https://doi.org/10.3168/JDS.2017-12814
  • Circannual changes in major chemical composition of bulk dromedary camel milk as determined by FT-MIR spectroscopy, and factors of variation. Food Chemistry, 278, 248–253. https://doi.org/10.1016/J.FOODCHEM.2018.11.059
  • Physico-chemical quality of camel milk. Journal of Agriculture and Social Sciences, 2, 164–166.
  • Laboratory analysis of milk and dairy products. In R. C. Chandan, A. Kilara, & N. P. Shah (Eds.), Dairy processing and quality assurance (pp. 600–646). John Wiley & Sons. https://doi.org/10.1002/9781118810279.CH24

Materials and Methods

L 110-111, Add stage of lactation

Reply: Added (lines 167-168) 

Results and Discussion

In general, results are only presented in the main text without proper discussion; you can add the below article and factors to support the discussion.

L176, add discussion

Reply: Added (line 230)

Section 3.2, support this section with the most recent meta-analysis study related to the effect of season, breed and country on camel milk composition

Camel milk composition by breed, season, publication year, and country: A global systematic review, meta-analysis, and meta-regression. Compr Rev Food Sci Food Saf. 2022 May;21(3):2520-2559.

Add to the discussion part, variations in camel milk composition are also due to several factors, including the number of samples, different analytical techniques, and livestock management.

Reply: The corresponding discussion was added (lines 361-372)

Throughout the manuscript, the use of the English language needs to be improved.

Reply: revised

Reviewer 2 Report

The ms. “Yield and composition variations of the milk from different camel breeds in Saudi Arabia” (Ms. Ref. No. sci-2083333-v1) presents original results on the systematic analysis of seasonal variability regarding the yield and composition of camel milk collected from Jeddah, Riyadh, and Alwagh regions of Saudi Arabia from Majahem, Safra, Wadha, and Hamra breeds.

The topic falls within the aims and scopes of the Sci journal and of the targeted section.

Originality = Fair, the ms. presents original work on the yield and composition of camel milk.

Applicability: There is a geographical restriction regarding the applicability of the results, limited to the regions where camels are grown.

However, there is a series of drawbacks that need to be addressed to increase the presentation of the results and the overall quality of the manuscript.

Major issues:

1.     The main drawback of the ms. is related to the fact that the authors have completely disregarded the fatty acids composition of the camel milk. It would be good if the authors could determine the fatty acid profiles (FAP) for the investigated milk samples. However, since the samples were collected some years ago, this might not be possible. Therefore, I advise them to present the fatty acids composition of the camel milk in the Introduction and to point out the differences compared to cow and sheep milk. If you do not determine the FAP yourself, then I suggest compiling data from several papers, to strengthen your work. Please find relevant information in excellent recent papers: https://doi.org/10.3390/foods10092158; https://doi.org/10.1016/j.lwt.2021.113036; https://doi.org/10.3390/foods9111722; https://doi.org/10.3390/foods11070926.

2.     Regarding the discussion of the proteins, it is worth pointing out that camel milk has been recently shown excellent to obtain hypoallergenic infant milk formulae and is consequently a good alternative to cow-milk based products, due to its specific composition (https://doi.org/10.1016/j.lwt.2021.112813; https://doi.org/10.3390/foods11070926).

3.     Lines 105-107: The authors should mention “camel milk”.

4.     Figure 1 does not brink any science advancements, as it only depicts four camel breeds. Therefore, it throws a negative light on the manuscript if it is kept in the spotlight. Consequently, I advise to give it as a Supplementary material. Please revise.

5.     Please discuss the proteomics with respect to recently published papers: https://doi.org/10.1016/j.lwt.2021.112287;  https://doi.org/10.1016/j.lwt.2020.110842; Also, do not only present the results, but also comment them and showcase the importance of the camel milk proteins. For example, certain proteins have potential antihypertensive properties (https://doi.org/10.1016/j.lwt.2021.111135).

6.     In the M&M section: How many replicates were analyzed for each sample and for each parameter?

7.     In the Discussion section: Milk fat is one of the most expensive commodities, and therefore may be subject to fraudulent practices. Therefore, I suggest the authors to point out how the obtained results may help in detecting adulterated products. For example, it was recently shown that the butyric acid content may be used as a marker to distinguish among fats from dairy or non-dairy origin, as well as to distinguish milk fats from different species. (10.3168/jds.2020-19883), as it determines lower values of the saponification values of the milk fats (https://doi.org/10.3390/foods11101466). Therefore, it is worth mentioning that based on these observations, a possible in-depth continuation of the study is to observe the variability of the examined breeds in terms of their short-chain fatty acids and especially the butyric acid, as a useful tool for camel milk products authentication or detection of fraud. Please revise.

8.     The Lactoscan device usually comes set up for cow or sheep milk analysis. How were the parameters adapted for the camel milk?

9.     Line 129: “almost was veterinary doctor or bachelor level person”. This portion needs rephrasing.

10.  I suggest performing two-way ANOVA on the data presented in Table 2, as there should be pointed out the variation across breeds and across seasons. Please revise and update the table with the letters for the statistical significance.

11.  The references seem quite old; I suggest updating with more recent references to showcase the actuality of the investigated subject.

12.  Please carefully revise the English.

Given the completed score sheet and the comments above, after careful evaluation, the ms. “Yield and composition variations of the milk from different camel breeds in Saudi Arabia” (Ms. Ref. No. sci-2083333-v1) needs MAJOR REVISION according to comments prior to consideration for publication in Sci journal.

Author Response

Reviewer #2

The ms. “Yield and composition variations of the milk from different camel breeds in Saudi Arabia” (Ms. Ref. No. sci-2083333-v1) presents original results on the systematic analysis of seasonal variability regarding the yield and composition of camel milk collected from Jeddah, Riyadh, and Alwagh regions of Saudi Arabia from Majahem, Safra, Wadha, and Hamra breeds.

The topic falls within the aims and scopes of the Sci journal and of the targeted section.

Originality = Fair, the ms. presents original work on the yield and composition of camel milk.

Applicability: There is a geographical restriction regarding the applicability of the results, limited to the regions where camels are grown.

However, there is a series of drawbacks that need to be addressed to increase the presentation of the results and the overall quality of the manuscript.

Major issues:

  1. The main drawback of the ms. is related to the fact that the authors have completely disregarded the fatty acids composition of the camel milk. It would be good if the authors could determine the fatty acid profiles (FAP) for the investigated milk samples. However, since the samples were collected some years ago, this might not be possible. Therefore, I advise them to present the fatty acids composition of the camel milk in the Introduction and to point out the differences compared to cow and sheep milk. If you do not determine the FAP yourself, then I suggest compiling data from several papers, to strengthen your work. Please find relevant information in excellent recent papers: https://doi.org/10.3390/foods10092158; https://doi.org/10.1016/j.lwt.2021.113036; https://doi.org/10.3390/foods9111722; https://doi.org/10.3390/foods11070926.

Reply: thank you for the reviewer on the comment and understanding the samples situation. Discussion of the importance of fatty acids and description of fatty acid profile in milk from other animals was added (lines 105-134).

  1. Regarding the discussion of the proteins, it is worth pointing out that camel milk has been recently shown excellent to obtain hypoallergenic infant milk formulae and is consequently a good alternative to cow-milk based products, due to its specific composition (https://doi.org/10.1016/j.lwt.2021.112813; https://doi.org/10.3390/foods11070926).

Reply: thank you for the reviewer on the advanced comment, the discussion of the allergic superiority of cow-milk in comparison with camel milk was added (lines 467-480)

  1. Lines 105-107: The authors should mention “camel milk”.

Reply: We mentioned it now (line 157) 

  1. Figure 1 does not brink any science advancements, as it only depicts four camel breeds. Therefore, it throws a negative light on the manuscript if it is kept in the spotlight. Consequently, I advise to give it as a Supplementary material. Please revise.

Reply: We respectfully disagree with this statement and do not understand why this figure might throw a negative light on the manuscript if it is kept in the spotlight. In fact, many readers are not familiar with the appearance of the four camel breeds analyzed in this study. Therefore, in our view, this image should be present in the main text, as it will attract attention of readers.

  1. Please discuss the proteomics with respect to recently published papers: https://doi.org/10.1016/j.lwt.2021.112287; https://doi.org/10.1016/j.lwt.2020.110842; Also, do not only present the results, but also comment them and showcase the importance of the camel milk proteins. For example, certain proteins have potential antihypertensive properties (https://doi.org/10.1016/j.lwt.2021.111135).

Reply: thank you for the comment, a discussion on this point was added (lines 480-486)

  1. In the M&M section: How many replicates were analyzed for each sample and for each parameter?

Reply: Added (lines 199-200)

  1. In the Discussion section: Milk fat is one of the most expensive commodities, and therefore may be subject to fraudulent practices. Therefore, I suggest the authors to point out how the obtained results may help in detecting adulterated products. For example, it was recently shown that the butyric acid content may be used as a marker to distinguish among fats from dairy or non-dairy origin, as well as to distinguish milk fats from different species. (3168/jds.2020-19883), as it determines lower values of the saponification values of the milk fats (https://doi.org/10.3390/foods11101466). Therefore, it is worth mentioning that based on these observations, a possible in-depth continuation of the study is to observe the variability of the examined breeds in terms of their short-chain fatty acids and especially the butyric acid, as a useful tool for camel milk products authentication or detection of fraud. Please revise.

Reply: The corresponding discussion was added (lines 361-372)

  1. The Lactoscan device usually comes set up for cow or sheep milk analysis. How were the parameters adapted for the camel milk?

Reply: thank you for the reviewer on the comment. Analysis was conducted according to the manufacturer manual after the Lactoscan was calibrated for the camel milk with the help of the company programmer. The corresponding information is added (Lines 194-195)

  1. Line 129: “almost was veterinary doctor or bachelor level person”. This portion needs rephrasing.

Reply: This portion was rephrased limes 183-184.

  1. I suggest performing two-way ANOVA on the data presented in Table 2, as there should be pointed out the variation across breeds and across seasons. Please revise and update the table with the letters for the statistical significance.

Reply: Actually, we analyzed each milk parameter across camel clans inside each season using one-way ANOVA. The main object was to detect differences between clans milk composition during each season. The differences between seasons came in second importance to us and we did only for detecting any odd values during seasons.

  1. The references seem quite old; I suggest updating with more recent references to showcase the actuality of the investigated subject.

Reply: We added updated references and included several recent references.

  1. Please carefully revise the English.

Reply: Revised

Round 2

Reviewer 1 Report

The authors have done a lot of work on the paper and made substantial improvements. The literature review and discussion are now improved. I believe the paper makes a worthwhile contribution and could be accepted for publication.

Author Response

The authors have done a lot of work on the paper and made substantial improvements. The literature review and discussion are now improved. I believe the paper makes a worthwhile contribution and could be accepted for publication.

Response: We are glad that this reviewer is satisfied by the revision and are thankful to this reviewer for high evaluation of our work.

Reviewer 2 Report

Comments on the ms. “Yield and composition variations of the milk from different camel breeds in Saudi Arabia” (Ms. Ref. No. sci-2083333-v2) – Revision 1

After the first review round, the authors have in general suitably addressed the reviewers’ comments. Proper changes have been made in the ms. according to suggestions and consequently, the ms. was considerably improved compared to its initial submission. In general, I agree with the modifications on the manuscript. However, in the current form, the manuscript cannot be accepted because there are still a number of inconsistencies between the information in the manuscript and the referenced papers that need to be addressed.

1.     Regarding comment #2: Line 480 (revised ms.): The information is also supported by https://doi.org/10.3390/foods11070926 and should be mentioned. Please revise.

2.     Regarding comment #4: I disagree with the authors’ rebuttal. All the scientists working in this topic (analysis of camel milk) certainly know a lot about the various camel breeds. A simple photo of those four breeds is therefore very common knowledge and should not be placed in the article main body. After all, the targeted journal (Sci) has a good reputation and a highly specialized audience. Besides, I have not indicated to simply remove the photos, I only asked you to place them in the Supplementary material section (with the indication in the text as “see Figure S1”). In this way, you let the most relevant results of your article in the spotlight (where they should be), and not catch the reader’s eye with not relevant pictures of camels. In this way, even if your article will be read by someone who does not know about the four breeds, they can find the picture in the Supplementary file.

3.     Regarding comment #7: Line 369 (revised ms.): I disagree with the cited paper. It does not refer to the butyric moiety. The correct reference is DOI: 3168/jds.2020-19883 (When detection of dairy food fraud fails: An alternative approach through proton nuclear magnetic resonance spectroscopy). Please update.

4.     Line 372 (revised manuscript). The cited reference does not support the idea in the manuscript. The correct reference is https://doi.org/10.3390/foods11101466. Please update.

Consequently, after careful examination of the revised ms. “Yield and composition variations of the milk from different camel breeds in Saudi Arabia" (revised title, Ms. Ref. No. sci-2083333-v2), my recommendation term is MINOR REVISION.

Author Response

After the first review round, the authors have in general suitably addressed the reviewers’ comments. Proper changes have been made in the ms. according to suggestions and consequently, the ms. was considerably improved compared to its initial submission. In general, I agree with the modifications on the manuscript. However, in the current form, the manuscript cannot be accepted because there are still a number of inconsistencies between the information in the manuscript and the referenced papers that need to be addressed.

Response: We are thankful to this reviewer for critical reading of the revised manuscript and for providing useful comments. We  addressed all the remaining concerns and amended manuscript accordingly. We hope that revised version became more suitable for publication.

  1. Regarding comment #2: Line 480 (revised ms.): The information is also supported by https://doi.org/10.3390/foods11070926 and should be mentioned. Please revise.

Response: Thank you for pointing this out. The recommended reference was added to the revised manuscript (it is reference #76).

  1. Regarding comment #4: I disagree with the authors’ rebuttal. All the scientists working in this topic (analysis of camel milk) certainly know a lot about the various camel breeds. A simple photo of those four breeds is therefore very common knowledge and should not be placed in the article main body. After all, the targeted journal (Sci) has a good reputation and a highly specialized audience. Besides, I have not indicated to simply remove the photos, I only asked you to place them in the Supplementary material section (with the indication in the text as “see Figure S1”). In this way, you let the most relevant results of your article in the spotlight (where they should be), and not catch the reader’s eye with not relevant pictures of camels. In this way, even if your article will be read by someone who does not know about the four breeds, they can find the picture in the Supplementary file.

Response: Although we still think that this image should be present in the main text, we decided to follow the recommendation of this reviewer and moved Figure 1 to supplementary materials. We are doing this to avoid the unnecessary delay with the processing of our manuscript, which would likely to happen if we would insist that the Figure 1 should be kept in the main text.

  1. Regarding comment #7: Line 369 (revised ms.): I disagree with the cited paper. It does not refer to the butyric moiety. The correct reference is DOI: 3168/jds.2020-19883 (When detection of dairy food fraud fails: An alternative approach through proton nuclear magnetic resonance spectroscopy). Please update.

Response: Thank you for pointing this out. The recommended reference was added to the revised manuscript (it is a new reference #65)

  1. Line 372 (revised manuscript). The cited reference does not support the idea in the manuscript. The correct reference is https://doi.org/10.3390/foods11101466. Please update.

Response: Thank you for pointing this out. The recommended reference was added to the revised manuscript (it is a new reference #66)